# Prevalence and correlates of intimate partner sexual violence among pregnant women in Napak district, Northeastern Uganda

**Godfrey Patrick Amodoi**[1], **Ivan Mugisha Taremwa**[2]*, **Joan Nakakande**[1,3], **Pardon Akugizibwe**[4,5], **Samuel Mugambe**[1,2], **Miisa Nanyingi**[1]

1 Faculty of Health Sciences, Uganda Martyrs University, Kampala, Uganda, 2 Institute of Allied Health Sciences, Clarke International University, Kampala, Uganda, 3 Department of Community Health and Research, AMBSO, Kampala, Uganda, 4 Institute of Public Health and Management, Clarke International University, Kampala, Uganda, 5 Faculty of Health Sciences, Mountains of the Moon University, Fort Portal, Uganda

* imugisha@ymail.com

**Data Availability Statement:** All relevant data are within the paper and are available without any restrictions.

## Abstract

Intimate partner sexual violence (IPSV) during pregnancy is of key reproductive health concern as it is associated with various risks linked to severe intrapartum complications. This study assessed the prevalence and the correlates of intimate partner sexual violence among pregnant women in Napak district, Northeastern Uganda. This was an analytical cross-sectional study conducted among 284 pregnant women who were obtained by systematic sampling in Napak district between November and December 2020. A structured questionnaire was used to collect the data, and this was analyzed using STATA version 15. The correlates of IPSV were determined at a multivariable level using a Poisson regression model with robust variance at the individual, relationship, and societal levels of the socio-ecological model. The study enrolled 284 pregnant women of whom, 65.5% were aged between 18 and 28 years and 62.0% were not formally educated. Also, 56% of the pregnant women had experienced IPSV during their current pregnancies, the most prevalent form (35.6%) being unwanted sexual advances. Factors of women being in their third or subsequent pregnancies, primiparity, women who rated their communication with their partners as low, being in a relationship for less than five years, lower age of the spouse, and a lower level of education (primary) among women showed a statistical association with IPSV. This study reports a high prevalence of IPSV during pregnancy, and it was associated with individual, relationship, and societal factors. Based on this, concerted efforts through sensitization on the dangers of IPSV are required to avert the practice.

## Introduction

The elimination of all forms of violence against the female gender forms a critical component of the Sustainable Development Goals (SDG), particularly SDG 5 target 2 [1, 2]. The SDG 5.2 seeks: "to eliminate all forms of violence against all women and girls in the public and private

**Funding:** The authors received no specific funding for this work.

**Competing interests:** The authors have declared that no competing interests exist.

spheres, including trafficking, sexual and other types of exploitation". Despite global concerted efforts to end female gender violence, this SDG target has only taken a human rights stance, yet Intimate Partner Sexual Violence (IPSV) is associated with devastating reproductive health ramifications [3, 4]. As per the definition of the World Health Organization, IPSV can take any of the five forms, namely; rape, unwanted sexual advances, denial to use contraception, forced abortion, or denial to use protective measures against sexually transmitted infections (STIs) [5]. IPSV is life-threatening as it increases the risk of antepartum depression, which is associated with premature rupture of membranes and preterm births [6, 7]. Also, the denial of the use of protective measures against sexually transmitted infections (STIs) has more delipidating health implications, including increased risk of urinary tract infections and STIs [8] and vertical transmission of Human Immunodeficiency Virus (HIV) from the mother to the unborn baby [9]. Moreover, rape during pregnancy is associated with vaginal trauma, pelvic floor dysfunction, dyspareunia, vaginismus, bacterial vaginosis, and vaginitis [3, 4, 10, 11]. These adverse outcomes are associated with high-risk pregnancies, intrapartum, and postpartum complications [12, 13].

Whereas global efforts (legal and policy perspectives) have shaped the measures to mitigate IPSV [14, 15], IPSV remains unacceptably high (up to 70%) in Africa [16]; and 10% to 30% prevalence in East Africa [17]. In Uganda, varied reports have indicated IPSV ranging from 27.8% [18] to 70.3% [19]. Moreover, sociocultural factors such as forced marriages [19] have been reported in the northeastern region where Napak district is located. In this region, marriage and childbearing decisions are entrenched in a ritualist attribute. For example, cultural practices such as courtship of wrestling which has been used over time for acceptance and readiness to marry potentiates the risk of forced marriage and IPSV [20]. In Napak district and the Karamoja region, the incidences of intimate partner sexual violence have reportedly been on the rise, and this has also been linked to numerous adverse obstetric outcomes. Moreover, there is a dearth of studies on the prevalence and correlates of intimate partner sexual violence among pregnant women in the Karamoja region. This study determined the prevalence and the correlates of intimate partner sexual violence among pregnant women in Napak district, Northeastern Uganda.

## Materials and methods

### Study design, site, and duration

The study used an analytical cross-sectional survey design that was based on a positivist approach, which states that "the truth is out and requires that numerical data be collected and analyzed prior to making deductions that may agree or disagree with the stated truth" [21]. The study was conducted in Napak district located in the Karamoja region of inter-clan and cross-border violent cattle raids located in Northeastern Uganda. Napak district is comprised of six sub-counties, namely; Iriiri, Lokopo, Lopei, Lotome, Matany, and Ngoleriet sub-counties. The district has one Private-Not-For-Profit (PNFP) hospital, five health center IIIs, and five health center IIs. The PNFP hospital and all health center IIIs provide maternal services including antenatal, maternity, and child care delivery services. The study was conducted from 30th November 2020 to 30th December 2020.

### Study population and size

The study population comprised pregnant women (16–49 years) who were between 8 to 36 weeks of gestation. The ANC visits period was guided by the Ministry of Health, Uganda in adherence to the World Health Organization recommendation of the eight antenatal care contacts with the health system during each pregnancy [22, 23]. Pregnant women within the first

trimester tend to think of being pregnant after noticeable pregnancy-related symptoms and serial missed menstrual periods, and this may coincide with the pregnancy confirmation test performed as part of the first-trimester antenatal assessment. On the other hand, week 36 of gestation is the last visit within the third trimester, and at this time, some expecting mothers in the fourth antenatal schedule (≥36 weeks) may experience gestational discomforts in preparation for labor.

The sample size was guided by the target population size in Napak district being less than 10,000 as assessed by the district health reports for the financial report 2020/2021. On this premise, a sample size formula derived by Daniel [24] that incorporates a population correction factor was used. Considering a proportion of 1082 pregnant women seeking antenatal care in Napak district, and correcting for non-response; a total of 284 pregnant women were considered for this study.

## Participant enrollment

The study enrolled pregnant women who were married and living with a husband/partner at the time of the study. The participants were informed about the study during the antenatal care health talk, and they were informed of the study. The study team approached each of the potential participants and sought consent to participate in the study. A participant who accepted was then taken to a private room and written consent was obtained. The participant was administered a questionnaire and a member of the study team was available to explain any inquiries by the participant. The filled questionnaire was cross-checked for completeness and filed.

## Sampling procedure

As Matany Hospital is attended by the majority of the pregnant women in the district; it was purposively sampled so as not to miss out on the relatively larger pool it serves. Then, the study used a simple random sampling for the health center IIIs. The sampling frame of pregnant women of 8 to 36 weeks of gestation. The study presumed that within this period, any incident IPV could be remembered by the pregnant woman. To achieve this, a lottery approach was used in which the names of the five health center IIIs were written on equal-sized pieces of paper, folded, and put in a box. They were ruffled and one picked after each ruffle without replacement. That was done until three of the five papers had been picked, following which they were unfolded to reveal the names of the health centers to be sampled. At the three health centre IIIs, pregnant women were non-randomly sampled until the allocated sample size per facility (50 pregnant women for each Health Centre III) was attained. On the other hand, one hundred and thirty-four pregnant women were enrolled from Matany Hospital. This approach provided a representative sample and minimized selection bias.

## Data collection tool and approach

The study used a structured interviewer-administered questionnaire (S1 Appendix). This was developed based on the existing literature [6, 12, 13, 16–20], and was reviewed by an expert on sexual gender based violence. The interviews were conducted on a one-on-one basis, with husbands excluded from participating to ensure candid responses. These interviews took place in a health facility setting during the ANC visits of eligible women with dedicated rooms provided to ensure privacy and confidentiality. A pilot testing of the questionnaire was done among 20 pregnant women at Moroto Regional Referral Hospital, Moroto district, and changes were made accordingly to ensure its suitability. Data collection was conducted by the

study team and supported by three trained research assistants proficient in the local language (Ngakarimojong).

## Study variables and measurement of variables

The dependent variable of this study was the experience of IPSV during pregnancy, which was indicated by incidence of any, or all of the following; unwanted sexual advances defined as 'the partner attempted to have sexual contact ignoring the woman's objections, rape was defined as 'the woman was physically forced to have sexual intercourse when she did not want to and denial to use STI preventive measures defines as 'the woman was refused to STI preventive measures such as condom when she needed to use one' [25]. Each of the three screening questions of IPSV was an independent indicator of its incidence, as was affirmed by pregnant women. A woman was said to have been sexually violated by her intimate partner if she reported/having had her spouse make sexual advances to her when she indicated that she was not interested, forced to have sexual intercourse when she did not want to (rape), or denied the use STI preventive measures in the past 2 or 8 months. According to WHO [25], sexual advances made by an intimate partner without the consent or against the will of the other person is a form of intimate partner violence.

The independent variables were individual, relationship, and community factors as per the suppositions of the socio-ecological model [26]. The individual factors assessed having and nature of any disabilities, gestation stage, history and specificity of any substance abuse since conception, history and form of intimate partner violence in any previous pregnancies, and communication with the spouse. The relationship characteristics focused on the age, religion, level of education, spouse employment status, spousal history of violence, socio-cultural norms such as bride price, substance abuse, decision-making in the home, and neighborhood attributes. The community factors included perceived community practices (including a right to have sex with a wife, irrespective of the circumstance at hand, sex consideration during pregnancy, and reporting of improper conduct related to sexual practices in the community). The responses were scored on a Likert scale of 'strongly agree, agree, undecided, disagree, and strongly disagree. These responses were dichotomized under 'yes' (strongly agree, agree) and 'no' (undecided, disagree, and strongly disagree).

## Data management and analysis

Questionnaires were manually checked for completeness, and the data therein was coded and entered into EpiInfo$^{TM}$ version 7.0 (*Centre for Disease Control and Prevention*, *Antlanta*, *US*) and exported to STATA version 15 (*STATA Corporation*, *College Station*, *TX*) for analysis. Exploratory data analysis was done to check missing values and outliers. This was supplemented with the analysis of descriptive frequencies, and looking at the outputs, to find out where the totals and frequencies tally. Then, univariate analysis was conducted to analyze the frequencies and percentages of each variable in the studied population. This was followed by bivariable analysis without adjustment for confounders, using a generalized linear model, and modified Poisson with robust variance given that the magnitude of IPSV obtained was more than 10% [27]. Bivariate analysis was carried out with independent variables to select the variables for multivariable analysis at a p-value ≤0.20. Multivariate analysis was then considered in which all variables that were found to be statistically significant at bivariate analysis were fitted. We identified the correlates of IPSV stopping when all the remaining variables were significant at a 5% (0.05) level of significance. Backward stepwise Likelihood Ratio (LR) was used to find the significant predictors of IPSV among pregnant women. Statistical significance at this

point was set at 5% (p<0.05), and the findings were reported using adjusted prevalence ratios along with their corresponding confidence intervals at 95%.

### Ethical considerations

The study obtained ethical approval from the research and ethics committee of The AIDS Support Organization (TASO), Kampala (TASOREC/057/2020/-UG-REC-009). Administrative permission was obtained from Napak district health officer, and each of the participating health facilities. For the adult participants, the study obtained written informed consent, while pregnant women under 18 years, are considered mature minors, and therefore they provided assent. By the research regulations in Uganda, such are allowed to offer informed consent solely, without seeking the parent or guardians' consent. This is in reference to the Uganda National Council of Science and Technology (UNCST) guidelines (available at: https://www.uncst.go.ug/details.php?option=smenu&id=13&Downloads.html) subsection 5.8, page 19.

Participation was voluntary and the study ensured maximum confidentiality considering the intricacy of the study topic. Also, the team was mindful of the anticipated emotional discomfort from those victimized. To such, psychosocial support in the form of confidential counseling and post-trauma support was offered by trained healthcare workers who were members of the study team, and referrals were considered to a local non-governmental based organization.

## Results

The study enrolled 284 pregnant women of whom, the majority (n = 186, 65.5%) were young women aged between 18 and 28 years and 62.0% (n = 176) were not formally educated. Further, 84.5% (n = 240) of the respondents were cohabiting, and 54.2% (n = 154) had been in a marital relationship for less than five years. The socio- demographic characteristics of the respondents are given in Table 1.

### Intimate partner sexual violence

There were 35.6% (n = 101) pregnant women who had had unwanted sexual advances made by their spouses since conception of their current pregnancies. Moreover, 16.9% (n = 48) of the respondents had been denied the chance to use STI protection measures. Also, 22.9% (n = 65) of the respondents had had forced sexual intercourse, as shown in Table 2.

### Bivariate logistic regression analysis of the correlates of IPSV among pregnant women in Napak district, Uganda

The different variables considered showed a varying degree of association as depicted by the bivariate logistic regression analysis (Table 3), and variables with a p-value ≤0.2 were then considered for the multivariate logistic regression model.

### Multivariate logistic regression analysis of the correlates of IPSV

Women in their second pregnancy had a 21.5% lower prevalence of IPSV (aPR = 0.785, 95% CI: 0.684–0.901, p = 0.029) than those in their third or subsequent pregnancy. Similarly, women who had one pregnancy (para-one or primipara) had a 25% lower prevalence of IPSV (APR = 0.750, 95% CI: 0.571–0.985, p = 0.039) than women who had none (nulliparity). More, women who rated their communication with their partners as high had a 45% lower prevalence of IPSV (aPR = 0.555, 95%CI: 0.379–0.810, p = 0.002) than those who rated it as low. Women in relationships lasting less than five years, on the other hand, had a 1.53 times higher

**Table 1. Shows socio- demographic characteristics of the pregnant women sampled in Napak district, Uganda.**

| Variable, n = 284 | Frequency (Percentage) |
| --- | --- |
| **Age (in full years)** | |
| Less than 18 | 8(2.8) |
| 18 to 28 | 186(65.5) |
| 29 to 39 | 86(30.3) |
| 40 to 49 | 4(1.4) |
| **Current marital status** | |
| Married | 44(15.5) |
| Cohabiting | 240(84.5) |
| **Duration in marital relationship** | |
| Less than five years | 154(54.2) |
| More than five years | 130(45.8) |
| **Religious denomination** | |
| Catholic | 202(71.1) |
| Christian (Pentecostal) | 77(27.1) |
| Muslim | 5(1.8) |
| **Formally educated** | |
| Yes | 108(38.0) |
| No | 176(62.0) |
| **Level of education** | |
| Primary | 64(59.3) |
| Secondary | 38(35.2) |
| Post-secondary | 6(5.6) |
| **Currently employed** | |
| Yes | 62(21.8) |
| No | 222(78.2) |

prevalence of IPSV when compared to those in relationships lasting more than five years. IPSV prevalence decreased with increasing age of the spouse from 18 to 28 years (aPR = 9.94, 95%CI: 1.477–66.85, p = 0.018); 29 to 39 years (aPR = 9.385, 95%CI: 1.398–14.963, p = 0.021);

**Table 2. Shows intimate partner sexual violence among pregnant women in Napak district, Uganda.**

| Variable | Frequency (Percentage) |
| --- | --- |
| **Unwanted sexual advances** | |
| Yes | 101(35.6) |
| No | 183(64.4) |
| **Denied chance to use measures of protection from STI** | |
| Yes | 48(16.9) |
| No | 236(83.1) |
| **Forced sexual intercourse** | |
| Yes | 65(22.9) |
| No | 219(77.1) |
| **Frequency of rape pregnancy** | |
| Once | 28(43.1) |
| Twice | 29(44.6) |
| Thrice | 6(9.2) |
| More than thrice | 2(3.1) |

**Table 3. Showing the bivariate logistic regression analysis of IPSV among pregnant women in Napak district, Uganda.**

| Variable (Category) | IPSV | | Unadjusted risk ratio (95% CI) | p-values | Adjusted odds ratio (95% CI) | P-values |
|---|---|---|---|---|---|---|
| | No (%) n = 126 | Yes (%) n = 156 | | | | |
| **Individual level** | | | | | | |
| **Religion** | | | | | | |
| Catholic | 83 (65.9) | 119 (75.3) | 1 | | | |
| Other (Pentecostal/Islam) | 43 (34.1) | 39 (24.7) | 0.63 (0.37–1.05) | **0.082** | 0.43(0.22–0.91) | **0.028** |
| **Level of education** | | | | | | |
| No education | 72(57.1) | 104 (65.8) | 1 | | | |
| Primary | 29(23.0) | 35(22.2) | 0.84(0.47–1.49) | 0.541 | 0.50(0.20–1.26) | 0.141 |
| Post-primary | 25(19.9) | 19(12.0) | 0.53(0.27–1.03) | **0.060** | 1.31(0.35–4.86) | 0.689 |
| **Trimester of pregnancy** | | | | | | |
| Second | 46(36.5) | 70(44.3) | 1 | | | |
| Third | 80(63.5) | 88(55.7) | 0.72(0.45–1.17) | 0.185 | 0.78(0.38–1.66) | 0.533 |
| **Pregnancy ever carried** | | | | | | |
| one | 20(15.9) | 33(20.9) | 1 | | | |
| Two | 25(19.8) | 47(29.8) | 1.14(0.54–2.38) | 0.720 | 4.17(0.67–25.75) | 0.124 |
| Three | 28(22.2) | 31(19.5) | 0.67(0.32–1.43) | 0.300 | 0.49(0.07–3.53) | 0.481 |
| More than three | 53(42.1) | 47(29.8) | 0.54(0.27–1.06) | **0.074** | 0.68(0.06–8.18) | 0.763 |
| **Number of living children** | | | | | | |
| one | 38(30.2) | 41(26.0) | 1 | | | |
| Two | 22(17.5) | 34(21.5) | 1.43(0.72–2.87) | 0.311 | 6.96(1.52–30.82) | **0.013** |
| Three | 7(5.6) | 14(8.9) | 1.85(0.68–5.08) | 0.231 | 3.6(0.81–16.38) | 0.091 |
| More than three | 41(32.5) | 29(18.3) | 0.66(0.34–1.25) | 0.202 | 5.12(0.55–48.05) | 0.153 |
| None | 18(14.2) | 40(25.3) | 2.06(1.01–4.19) | **0.046** | 7.40(1.25-43-67) | **0.027** |
| **Was pregnancy planned** | | | | | | |
| Yes | 71(56.4) | 77(48.7) | 1 | | | |
| No | 55(43.7) | 81(51.3) | 1.36(0.85–2.17) | **0.200** | 0.54(0.23–1.23) | 0.147 |
| **Relationship level** | | | | | | |
| **Age of partner (years)** | | | | | | |
| 18–28 | 35 (27.8) | 57(36.1) | 1 | | | |
| 29–39 | 54(42.9) | 78(49.4) | 0.89(0.51–1.53) | 0.666 | 1.07(0.42–2.71) | 0.880 |
| 40–49 | 22(17.5) | 22(13.9) | 0.61(0.30–1.27) | 0.188 | 0.22(0.05–0.94) | **0.041** |
| >49 years | 15(11.9) | 1(0.6) | 0.04(0.01–0.32) | **0.002** | 0.01(0.001–0.112) | **< .001** |
| **Education level of partner** | | | | | | |
| No education | 61(48.4) | 100 (63.3) | 1 | | | |
| Primary level | 18(14.3) | 30(19.0) | 1.02(0.52–1.98) | 0.961 | 0.87(0.34–2.25) | 0.781 |
| Post-primary | 47(37.3) | 28(17.7) | 0.36(0.21–0.64) | **< .001** | 0.23(0.08–0.64) | **0.005** |
| **Partner employed** | | | | | | |
| Yes | 71(56.3) | 74(47.4) | 1 | | | |
| No | 56(43.7) | 47(52.6) | 1.43(0.89–2.29) | **0.137** | 2.82(1.28–6.17) | **0.010** |
| **Duration in the relationship** | | | | | | |
| < 5 years | 53 (42.1) | 101 (63.9) | 1 | | | |

*(Continued)*

**Table 3.** (Continued)

| Variable (Category) | IPSV | | Unadjusted risk ratio (95% CI) | p-values | Adjusted odds ratio (95% CI) | P-values |
|---|---|---|---|---|---|---|
| | No (%) n = 126 | Yes (%) n = 156 | | | | |
| ≥ 5years | 73 (57.9) | 57 (36.1) | 0.41 (0.25–0.66) | < .001 | 0.38(0.11–1.33) | 0.130 |
| Have you ever experienced IPSV when not pregnant? | | | | | | |
| No | 115 (91.3) | 92(58.2) | 1 | | | |
| Yes | 11(8.7) | 66(41.8) | 2.66(3.74–5.02) | < .001 | 11.06(3.77–35.72) | < .001 |
| **Level communication with partner** | | | | | | |
| High | 56(44.4) | 34(21.8) | | | | |
| Moderate | 63(50.0) | 106 (68.0) | 2.77(1.63–4.70) | < .001 | 2.83(1.32–6.04) | **0.006** |
| Low | 7(5.6) | 16(10.2) | 3.76(1.41–0.08) | **0.008** | 0.38(0.06–2.26) | 0.288 |
| **Partner involvement in gambling** | | | | | | |
| Yes | 15(11.9) | 38(24.1) | 1 | | | |
| No | 111 (88.1) | 120 (75.9) | 0.43(0.22–0.81) | **0.010** | 0.44(0.17–1.20) | 0.111 |
| **Partner allows visits to friends and neighbors** | | | | | | |
| Yes | 120 (95.3) | 144 (91.1) | 1 | | | |
| No | 6(4.8) | 14(8.9) | 1.94(o.72-5.21) | **0.186** | 0.76(0.13–4.34) | 0.761 |
| **Societal level** | | | | | | |
| **Partner has gender preferences for children** | | | | | | |
| Yes | 14(11.1) | 29(18.4) | 1 | | | |
| No | 112 (88.9) | 129 (81.7) | 0.55(0.28–1.10) | **0.094** | omitted | |
| **Gender preference(n = 43)** | | | | | | |
| Female | 10(71.4) | 9(31.0) | 1 | | | |
| Male | 4(28.6) | 20(69.0) | 5.56(1.36–22.56) | **0.016** | 5.56(1.36–22.56) | **0.016** |
| **Marital rape is not considered criminal** | | | | | | |
| No | 61(48.4) | 59(37.3) | 1 | | | |
| Yes | 65(51.6) | 99(62.7) | 1.57(0.98–2.53) | **0.061** | 1.38(0.83–2.29) | 0.216 |
| | | | | | | |
| **Does anyone provide emotional support?** | | | | | | |
| No | 30(23.8) | 21(13.3) | 1 | | | |
| Yes | 96(76.2) | 137 (86.7) | 2.04(1.10–3.77) | 0.023 | 1.80(0.94–3.46) | 0.077 |
| A pregnant woman sexually abused is not allowed to report (family elder, local council authorities, police) | | | | | | |
| No | 26(20.6) | 18(11.4) | | | | |
| Yes | 100 (79.4) | 140 (88.6) | 2.02(1.05–3.89) | **0.035** | 1.93(0.95–3.89) | 0.065 |

and 40 to 49 years (aPR = 7.954, 95%CI:1.165–54.286, p = 0.034) compared to older spouses over 49 years. Also, spouses with a higher level of education (secondary) had a 30% lower prevalence of IPSV (aPR = 0.669, 95%CI: 0.537–0.833, p = 0.001), while women with a lower level of education (primary) had a 2.93 higher prevalence of IPSV (aPR = 2.927, 95%CI: 1.211–7.075, p = 0.017) than those with a higher level of education (secondary), as shown in Table 4.

**Table 4. Showing the multivariable logistic regression analysis of IPSV among pregnant women in Napak district, Uganda.**

| Variable | Pregnancy IPSV | | cPR (CI at 95%) | P value | aPR (CI at 95%) | P value |
|---|---|---|---|---|---|---|
| | Experienced n = 158 | Not experienced n = 126 | | | | |
| **Gravidity** | | | | | | |
| One | 33(62.3%) | 20(37.7%) | 1.325 (0.986–1.780) | 0.062 | 0.813 (0.703–0.941) | 0.066 |
| Two | 47(65.3%) | 25(34.7%) | 1.389 (1.063–1.815) | **0.016** | 0.785 (0.684–0.901) | **0.029** |
| Three | 31(52.5%) | 28(47.5%) | 1.118 (0.812–1.539) | 0.494 | 0.894 (0.787–1.015) | 0.498 |
| More than three | 47(47.0%) | 53(53.0%) | 1.000 | | 1.000 | |
| **Parity** | | | | | | |
| One | 41(51.9%) | 38(48.1%) | 0.753 (0.572–0.989) | **0.042** | 0.750 (0.571–0.985) | **0.039** |
| Two | 34(60.7%) | 22(39.3%) | 0.880 (0.670–1.156) | 0.359 | 0.874 (0.658–1.161) | 0.353 |
| Three | 14(66.7%) | 7(33.3%) | 0.967 (0.682–1.369) | 0.849 | 0.980 (0.689–1.393) | 0.910 |
| More than three | 29(41.4%) | 41(58.6%) | 0.601 (0.433–0.834) | **0.002** | 0.599 (0.426–0.843) | **0.003** |
| None | 40(69.0%) | 18(31.0%) | 1.000 | | 1.000 | |
| **Level of communication with spouse** | | | | | | |
| High | 34(37.8%) | 56(62.2%) | 0.543 (0.372–0.793) | **0.002** | 0.555 (0.379–0.810) | **0.002** |
| Moderate | 106(62.7%) | 63(37.3%) | 0.902 (0.672–1.210) | 0.490 | 0.926 (0.687–1.249) | 0.616 |
| Low | 16(69.6%) | 7(30.4%) | 1.000 | | 1.000 | |
| **Duration in marital relationship** | | | | | | |
| Less than five years | 101(65.6%) | 53(34.4%) | 1.496 (1.194–1.874) | **<0.001** | 1.533 (1.210–1.941) | **<0.001** |
| More than five years | 57(43.8%) | 73(56.2%) | 1.000 | | 1.000 | |
| **Age range of spouse** | | | | | | |
| Between 18 and 28 years | 57(62.0%) | 35(38.0%) | 9.913 (1.476–66.574) | **0.018** | 9.944 (1.477–66.950) | **0.018** |
| Between 29 and 39 years | 78(59.1%) | 54(40.9%) | 9.455 (1.410–63.404) | **0.021** | 9.383 (1.398–14.963) | **0.021** |
| Between 40 and 49 years | 22(50.0%) | 22(50.0%) | 8.000 (1.172–54.600) | **0.034** | 7.954 (1.165–54.286) | **0.034** |
| More than 49 years | 1(6.3%) | 15(93.8%) | 1.000 | | 1.000 | |
| **Spouse formally educated** | | | | | | |
| Yes | 58(47.2%) | 65(52.8%) | 0.759 (0.608–0.948) | **0.015** | 0.669 (0.537–0.833) | **<0.001** |
| No | 100(62.1%) | 61(37.9%) | 1.000 | | 1.000 | |
| **Level of education of woman** | | | | | | |
| Primary | 30(62.5%) | 18(37.5%) | 2.656 (1.097–6.433) | **0.030** | 2.927 (1.211–7.075) | **0.017** |
| Secondary | 24(41.4%) | 34(58.6% | 1.759 (0.708–4.369) | 0.224 | 1.662 (0.668–4.134) | 0.274 |
| Post-secondary | 4(23.5%) | 13(76.5%) | 1.000 | | 1.000 | |

## Discussion

IPSV arguably poses a subtle challenge to reproductive health as it attributes to various life-threatening complications, namely; postpartum hemorrhage, sepsis, complications of childbirth, and fetomaternal mortality [2]. In this study, the prevalence of IPSV was 56% (n = 158), and this was higher compared to most previous studies, for example 40.8% reported from northern Ethiopia [16], 19.9% in western Ethiopia [28], 44.6% in Nigeria [29], 18.8% in Tanzania [30], 9.2% [31] and 35.3% [32] in Kenya, and 10.6% reported from a previous Ugandan study [18]. On the other hand, the obtained prevalence is lower than the 70.3% reported from southwestern Uganda [19]. The variations in the reported IPSV attest to its growing and disproportionate burden for varied ethnicities. Contrasting to the majority of the studies, this study considered IPSV as defined by the World Health Organization, which includes rape, unwanted sexual advances, denial to use contraception, forced abortion, or denial to use protective measures against sexually transmitted infections (STIs) [5]. Also, the studied population is highly patriarchal, and this socio-traditional fabric may as well account for the observed

high IPSV prevalence. This suggests an urgent need for concerted ethnical approaches to avert the sociocultural dynamics that negates reproductive health outcomes.

The study indicated that the prevalent form of IPSV was unwanted sexual advances (N = 101, 35.6%), a finding that may suggest no physical, but rather emotional violation. Consistent with previous studies, the associated violence may complicate the gestational process as well as augmenting the parturition [3, 4, 6, 7].

As the prevalence of IPSV was higher in multigravidas, both the physiological and anatomic variances may be considered. It seems multigravidas tend to experience more pregnancy-associated mood changes and may impair sexual drive. Premised on this, multigravidas tend to be less receptive to their spouse's sexual advances and are thus martially raped. Also, whereas sexual drive reduces in the first and third trimesters among all women; the effect is more exacerbated among multigravida women as they additionally experience other pregnancy complications [33]. Relationships for less than five years implied that perhaps the couple had yet cultivated healthy communication between them, which makes dialogue-less likely, thus increasing the risk of IPSV. This is consistent with previous reports [34, 35], and emphasizes relationship gaps due to breakdown in communication and negotiations. The age of the spouse was significantly correlated with IPSV, consistent with previous reports [36–38]. Young age translated to more IPSV, and this may relate both to hormonal and behavioral attributes. Also, with unemployability and emerging drug as well as substance abuse, it is plausible to find a high IPSV in this age range. From this study, education lessened the risk of IPSV. Consistent with other studies [34, 37], formal education comes with numerous positive attributes including being employed and also possible knowledge about the effects of IPSV. Generally, educated men tend to have relatively less controlling behavior due to the limited or non-embracement of patriarchy and male dominance.

The findings of this study ought to be interpreted in light of some limitations. IPSV was self-reported and this depends on the recall and respondents ability to disclose a rather confidential marital prospect. There are population-specific traditional norms that may affect the study findings. The study did not qualitatively explore the practice of IPSV, and this leaves a knowledge gap that requires critical attention to eliminate the IPSV practice. Also, this study may be limited by the interviewer bias and social desirability responses. More, this study reported on partners'/men's characteristics such as their involvement in risky behavior like alcohol/drug abuse, which would be of great value to understanding the IPSV practice and informing policy.

## Conclusion

This study reports a high prevalence of IPSV among pregnant women and it was associated with women being in their third or subsequent pregnancies, nulliparity, poor communication with a partner, being in a relationship for less than five years, lower age of the spouse, and a lower level of education. Based on the findings, it is important to sensitize the public on the dangers of IPSV to avert the practice required. A multipronged approach to health education and male involvement in addressing the IPSV practice is desired.

## Supporting information

**S1 Appendix. Data collection tool.**
(DOCX)

## Acknowledgments

We are grateful to the study participants and research assistants.

## Author Contributions

**Conceptualization:** Godfrey Patrick Amodoi, Ivan Mugisha Taremwa, Joan Nakakande, Pardon Akugizibwe, Samuel Mugambe, Miisa Nanyingi.

**Data curation:** Godfrey Patrick Amodoi, Ivan Mugisha Taremwa, Joan Nakakande, Pardon Akugizibwe, Samuel Mugambe, Miisa Nanyingi.

**Formal analysis:** Godfrey Patrick Amodoi, Ivan Mugisha Taremwa, Joan Nakakande, Pardon Akugizibwe, Samuel Mugambe, Miisa Nanyingi.

**Investigation:** Ivan Mugisha Taremwa, Joan Nakakande.

**Methodology:** Godfrey Patrick Amodoi, Ivan Mugisha Taremwa, Joan Nakakande, Pardon Akugizibwe, Samuel Mugambe, Miisa Nanyingi.

**Project administration:** Godfrey Patrick Amodoi, Ivan Mugisha Taremwa.

**Resources:** Ivan Mugisha Taremwa.

**Supervision:** Ivan Mugisha Taremwa, Joan Nakakande, Pardon Akugizibwe, Samuel Mugambe, Miisa Nanyingi.

**Writing – original draft:** Godfrey Patrick Amodoi, Ivan Mugisha Taremwa, Joan Nakakande, Pardon Akugizibwe, Samuel Mugambe, Miisa Nanyingi.

**Writing – review & editing:** Godfrey Patrick Amodoi, Ivan Mugisha Taremwa, Joan Nakakande, Pardon Akugizibwe, Samuel Mugambe, Miisa Nanyingi.

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
