## [Decision Letter · Decision Letter 0]

20 Sep 2023

PGPH-D-23-01429

Prevalence and Correlates of Intimate Partner Sexual Violence among Pregnant Women in Napak District, Northeastern Uganda

Dear Dr. Taremwa,

Thank you for submitting your manuscript to PLOS Global Public Health. After careful consideration, we feel that it has merit but does not fully meet PLOS Global Public Health’s publication criteria as it currently stands. Therefore, we invite you to submit a revised version of the manuscript that addresses the points raised during the review process.

Kindly respond to reviewers feedback, especially the points raised on methodology and discussion section. Clarifying these points will benefit the readers.

We look forward to receiving your revised manuscript.

Kind regards,

Muthusamy Sivakami

Academic Editor

Journal Requirements:

1. We have noticed that you have uploaded Supporting Information files, but you have not included a list of legends. Please add a full list of legends for your Supporting Information files after the references list. 

Additional Editor Comments (if provided):

Kindly respond to our reviewers, especially reviewer 1. We appreciate your interest with PLOS-GPH.

Reviewers' comments:

Reviewer's Responses to Questions

**Comments to the Author**

1. Does this manuscript meet PLOS Global Public Health’s publication criteria? Is the manuscript technically sound, and do the data support the conclusions? The manuscript must describe methodologically and ethically rigorous research with conclusions that are appropriately drawn based on the data presented.

Reviewer #1: Yes

Reviewer #2: Yes

2. Has the statistical analysis been performed appropriately and rigorously?

Reviewer #1: Yes

Reviewer #2: Yes

3. Have the authors made all data underlying the findings in their manuscript fully available (please refer to the Data Availability Statement at the start of the manuscript PDF file)?

Reviewer #1: No

Reviewer #2: Yes

4. Is the manuscript presented in an intelligible fashion and written in standard English?

Reviewer #1: Yes

Reviewer #2: Yes

5. Review Comments to the Author

Reviewer #1: Great work. This is a very important subject and will definitely contribute to prevention of IPSV.

It could even be better by attending to reviewer comments. Please do not assume that the your reader will understands everything in the methods section. There are parts that are a bit unclear and need to be clarified to improve the quality of this paper.

Reviewer #2: A good attempt by the authors. The issue of IPSV during pregnancy is a very serious and important RH topic, particularly in a society or sub-population experiencing such high prevalence (56%). However, the paper could have elaborated on the forms of 'unwanted sexual advances' and circumstances leading to 'rape or forced sexual intercourse'. Some may argue that the partners/men are unaware of their partners' pregnancy status (first trimester, etc) or ignorant of reproductive rights. Inclusion of some of partners'/men's characteristics in the analysis (if data permits; e.g. their age, education, economic status, risky behaviours like alcohol/drug abuse, gambling, etc), which could be of great value to policy and our understanding of such issue.

Authors are advised to cross-check the prevalent values (%) mention in the text with that in the relevant tables. For instance, in Discussion section, page 12 (last para; lines 226-227), on unwanted sexual advances the authors wrote, N=100, 63.3%, which is different from the values and number in the Table 2 and in the text in page 9, line 187.

6. PLOS authors have the option to publish the peer review history of their article (what does this mean?). If published, this will include your full peer review and any attached files.

**Do you want your identity to be public for this peer review?** For information about this choice, including consent withdrawal, please see our Privacy Policy.

Reviewer #1: **Yes: **Malachi Arunda

Reviewer #2: No

---

## [Decision Letter · Decision Letter 1]

6 Dec 2023

PGPH-D-23-01429R1

Prevalence and Correlates of Intimate Partner Sexual Violence among Pregnant Women in Napak District, Northeastern Uganda

Dear Dr. Taremwa,

Thank you for submitting your manuscript to PLOS Global Public Health. After careful consideration, we feel that it has merit but does not fully meet PLOS Global Public Health’s publication criteria as it currently stands. Therefore, we invite you to submit a revised version of the manuscript that addresses the points raised during the review process.

Our reviewer has flagged some more important points, especially in the methodology section and discussion. Kindly amend them the earliest, which would help us move to next stage. Good luck.

We look forward to receiving your revised manuscript.

Kind regards,

Muthusamy Sivakami

Academic Editor

Journal Requirements:

Additional Editor Comments (if provided):

Our reviewer has flagged some more important points, especially in the methodology section and discussion. Kindly amend them the earliest, which would help us move to next stage. Good luck.

Reviewers' comments:

Reviewer's Responses to Questions

**Comments to the Author**

1. If the authors have adequately addressed your comments raised in a previous round of review and you feel that this manuscript is now acceptable for publication, you may indicate that here to bypass the “Comments to the Author” section, enter your conflict of interest statement in the “Confidential to Editor” section, and submit your "Accept" recommendation.

Reviewer #3: (No Response)

2. Does this manuscript meet PLOS Global Public Health’s publication criteria? Is the manuscript technically sound, and do the data support the conclusions? The manuscript must describe methodologically and ethically rigorous research with conclusions that are appropriately drawn based on the data presented.

Reviewer #3: Yes

3. Has the statistical analysis been performed appropriately and rigorously?

Reviewer #3: I don't know

4. Have the authors made all data underlying the findings in their manuscript fully available (please refer to the Data Availability Statement at the start of the manuscript PDF file)?

Reviewer #3: Yes

5. Is the manuscript presented in an intelligible fashion and written in standard English?

Reviewer #3: No

6. Review Comments to the Author

Reviewer #3: This issue is indeed a significant public health concern, and limited evidence is available globally.

Overall, writing needs to be improved.

Introduction

It needs improvement, and the rationale for the study needs better articulation.

In some places, too many things in a single sentence make it difficult to read and comprehend.

Method

Sections need more clarity on the basis on which authors decided to select 3 out of 5 centres

Elaborate on the process of participant recruitment.

The table 3 & 4 headings are not consistent with the type of analysis mentioned in the method section.

In many places, the CI was set differently, e.g., 0.20 or 0.05. There should be a consistent way to report such information.

Results and Discussion

The interpretation of the results needs a careful approach.

The discussion part mentions so many things, but as a reader, it is most likely to get lost. For example, in many places, authors say that things are due to patriarchal systems and cultural norms. on the other hand, they also try to say that these are due to biological changes, etc. The reviwer is aware that there can be multiple explanations of the findings, but these should be sync and consistent.

References:

A few references mentioned are not valid or accessible. For example, the WHO acts on IPSV. Another example is the prevalence of IPSV.

Note: Details comments are mentioned in the enclosed document.

7. PLOS authors have the option to publish the peer review history of their article (what does this mean?). If published, this will include your full peer review and any attached files.

**Do you want your identity to be public for this peer review?** For information about this choice, including consent withdrawal, please see our Privacy Policy.

Reviewer #3: No

---

## [Decision Letter · Decision Letter 2]

4 Jan 2024

Prevalence and Correlates of Intimate Partner Sexual Violence among Pregnant Women in Napak District, Northeastern Uganda

PGPH-D-23-01429R2

Dear Mr. Taremwa,

We are pleased to inform you that your manuscript 'Prevalence and Correlates of Intimate Partner Sexual Violence among Pregnant Women in Napak District, Northeastern Uganda' has been provisionally accepted for publication in PLOS Global Public Health.

Best regards,

Muthusamy Sivakami

Academic Editor

Thank your for addressing all the feedback. It is great to see the revised paper.

Reviewer Comments (if any, and for reference):

Reviewer's Responses to Questions

**Comments to the Author**

1. If the authors have adequately addressed your comments raised in a previous round of review and you feel that this manuscript is now acceptable for publication, you may indicate that here to bypass the “Comments to the Author” section, enter your conflict of interest statement in the “Confidential to Editor” section, and submit your "Accept" recommendation.

Reviewer #3: All comments have been addressed

2. Does this manuscript meet PLOS Global Public Health’s publication criteria? Is the manuscript technically sound, and do the data support the conclusions? The manuscript must describe methodologically and ethically rigorous research with conclusions that are appropriately drawn based on the data presented.

Reviewer #3: Yes

3. Has the statistical analysis been performed appropriately and rigorously?

Reviewer #3: Yes

4. Have the authors made all data underlying the findings in their manuscript fully available (please refer to the Data Availability Statement at the start of the manuscript PDF file)?

Reviewer #3: Yes

5. Is the manuscript presented in an intelligible fashion and written in standard English?

Reviewer #3: Yes

6. Review Comments to the Author

Reviewer #3: All comments have been addressed adequately.

7. PLOS authors have the option to publish the peer review history of their article (what does this mean?). If published, this will include your full peer review and any attached files.

**Do you want your identity to be public for this peer review?** For information about this choice, including consent withdrawal, please see our Privacy Policy.

Reviewer #3: No
